# An Array of On-Chip Integrated, Individually Addressable Capacitive Field-Effect Sensors with Control Gate: Design and Modelling

**DOI:** 10.3390/s21186161

**Published:** 2021-09-14

**Authors:** Arshak Poghossian, Rene Welden, Vahe V. Buniatyan, Michael J. Schöning

**Affiliations:** 1MicroNanoBio, Liebigstr. 4, 40479 Düsseldorf, Germany; 2Institute of Nano- and Biotechnologies (INB), FH Aachen, Campus Jülich, Heinrich-Mußmannstr. 1, 52428 Jülich, Germany; welden@fh-aachen.de; 3Laboratory for Soft Matter and Biophysics, KU Leuven, Celestijnenlaan 200D, 3001 Leuven, Belgium; 4Department of Microelectronics and Biomedical Devices, National Polytechnic University of Armenia (NPUA), 105 Teryan St., NPUA, Yerevan 0009, Armenia; vbuniat@seua.am; 5Institute of Biological Information Processing (IBI-3), Forschungszentrum Jülich, 52425 Jülich, Germany

**Keywords:** capacitive field-effect sensor, on-chip integrated addressable EISCAP sensors, control gate, multianalyte detection, modelling, equivalent circuit

## Abstract

The on-chip integration of multiple biochemical sensors based on field-effect electrolyte-insulator-semiconductor capacitors (EISCAP) is challenging due to technological difficulties in realization of electrically isolated EISCAPs on the same Si chip. In this work, we present a new simple design for an array of on-chip integrated, individually electrically addressable EISCAPs with an additional control gate (CG-EISCAP). The existence of the CG enables an addressable activation or deactivation of on-chip integrated individual CG-EISCAPs by simple electrical switching the CG of each sensor in various setups, and makes the new design capable for multianalyte detection without cross-talk effects between the sensors in the array. The new designed CG-EISCAP chip was modelled in so-called floating/short-circuited and floating/capacitively-coupled setups, and the corresponding electrical equivalent circuits were developed. In addition, the capacitance-voltage curves of the CG-EISCAP chip in different setups were simulated and compared with that of a single EISCAP sensor. Moreover, the sensitivity of the CG-EISCAP chip to surface potential changes induced by biochemical reactions was simulated and an impact of different parameters, such as gate voltage, insulator thickness and doping concentration in Si, on the sensitivity has been discussed.

## 1. Introduction

Biosensors for multianalyte detection attracted much attention in many fields of application, including point-of-care and clinical diagnostics, food and drug screening, environmental monitoring, etc. Electrolyte-gated field-effect devices (EG-FED) have been recognized as a promising transducer in designing chemical and biological sensors because of their small size and weight, fast response time, real-time monitoring, label-free and multiplexed biomolecular detection, possibility of on-chip integration of EG-FEDs and signal-processing circuit and compatibility to micro- and nanofabrication technologies with the future prospect of large-scale production at relatively low costs [1,2,3,4,5,6,7,8]. In addition, miniaturized analysis systems (e.g., lab-on-a-chip devices or electronic tongues) based on on-chip integrated EG-FEDs in an array format have received tremendous attention due to their ability for multiplexed and (quasi)simultaneous assaying of multiple chemical or biological species [9,10,11,12,13,14]. Such multiplexed biochemical sensing systems may offer several advantages over devices for single-analyte detection, such as reduced assay time and sample volume, reduced costs and high throughput.

The electrolyte-insulator-semiconductor capacitor (EISCAP) belongs to the family of EG-FEDs and represents a biochemically sensitive capacitor [15]. In contrast to ISFETs (ion-sensitive field-effect transistor) or Si nanowire transistors, EISCAPs have a simple structure (see Figure 1a) and are easy and low-cost in fabrication; typical preparation steps do not require photolithographic patterning, and the implementation of a simple O-ring provides sufficient protection of the conductive regions of the EISCAP from the electrolyte solution. At the same time, the results achieved with EISCAPs are fully transferable to other EG-FEDs, thereby circumventing the need for fabrication of complicated transistor structures. At present, a lot of single EISCAP sensors modified with particular recognition elements have been developed and successfully proved for the detection of pH [16], concentration of ions [17], enzyme-substrate reactions [18,19,20,21], charged biomolecules (nucleic acids, proteins, biomarkers, nanoparticle/molecule hybrids) [22,23,24,25,26,27,28,29], plant virus particles [30], as well as for realizing biomolecular logic gates [31,32,33]. For recent progress in research and development of chemical sensors and biosensors based on EISCAPs, see [15].

In spite of successful experiments with single EISCAP sensors, however, the on-chip integration of multiple EISCAPs for multiplexed detection of multiple target analytes seems to be problematic, challenging the fabrication of electrically isolated, individually addressable capacitive structures: EISCAPs prepared on the same Si chip will stay interconnected via the common Si substrate. This may result in an unwanted cross-talk between the different EISCAPs in the array, thereby limiting the possibility to realize on-chip integrated multisensor systems. Only a few studies addressed this task in the literature. For example, Taing realized an EISCAP sensor array fabricated on a Si wafer that is anodically bonded to a glass substrate [34]. To obtain separate electrically decoupled EISCAPs, the Si wafer was diced by means of a saw cutter and, subsequently, the edges of the separated EISCAP chips were protected from contact with solution using a photoresist layer as schematically shown in Figure 1b. Another approach was proposed in [23], where an array of individually addressable nanoplate EISCAPs for chemical/biological sensing was developed using a SOI (silicon-on-insulator) wafer (Figure 1c). The nanoplate EISCAPs were prepared on a thin top Si layer (two photolithographic steps were needed). For isolation of the individual nanoplate capacitors, the top Si layer was anisotropically etched using the patterned top SiO_2_ layer as a mask. However, due to the large series lateral resistance of the top nanoplate Si, the frequency-dependent *C–V* curves of the nanoplate EISCAPs were deformed; they significantly differed from typical *C–V* plots of conventional EISCAPs [35]. Finally, a 2 × 2 array of on-chip integrated EISCAPs was demonstrated in [36], where the gate area of each sensor was separated by means of fabrication of individual electrolyte reservoirs, schematically illustrated in Figure 1d. Each EISCAP was addressed through an individual Au pseudo-reference electrode integrated onto the chip, which induced a large drift and instable sensor signal.

The above discussed examples demonstrate the possibility of realization of on-chip integrated EISCAPs. However, the price to be paid was the loss of the substantial advantages of EISCAP devices—their simple layout, as well as easy and cost-efficient preparation. In this work, we present a new and simple design, as well as the operational setup for an array of on-chip integrated, individually electrically addressable EISCAPs with a so-called control gate (CG) (further referred to as CG-EISCAP) as an alternative transducer structure for the multiplexed (quasi)simultaneous detection of multiple analytes without cross-talk effect between the individual sensors.

## 2. Design of On-Chip Integrated, Individually Addressable CG-EISCAPs

Figure 2 shows the schematic structure of the new designed sensor chip for detecting of multiple analytes, exemplarily combining three individual electrically addressable CG-EISCAPs (CG-EISCAP-1, CG-EISCAP-2 and CG-EISCAP-3). In comparison to conventional EISCAPs, which are based on an electrolyte-insulator-semiconductor system, the new designed CG-EISCAPs are composed of an electrolyte-insulator-metal-insulator-semiconductor structure. Here, the patterned metal layer (e.g., Au, Al) between the two insulators (insulator-1 and insulator-2) plays the role of the particular CG, in addition to the sensing gate (SG) using the common reference electrode (RE), similar to CMOS (complementary metal-oxide-semiconductor) floating- and programmable-gate ISFETs [37,38]. In order to protect the CG from contact with solution, it is covered with the top insulator-2 (e.g., Al_2_O_3_, Ta_2_O_5_) or stacked insulators (e.g., SiO_2_-Si_3_N_4_), which may also serve as a biochemical sensing layer (e.g., being pH-sensitive). In addition, the surface areas (spots) of insulator-2 above the CGs can be modified with various recognition elements (ionophores, enzymes, antibodies, nucleic acids, etc.), thereby making the CG-EISCAP chip sensitive to multiple analytes. Both insulator layers are assumed to be ideal, that is, no current passes through the insulator. For the measurement, the RE (e.g., a conventional Ag/AgCl RE) should provide a stable potential independent of pH or concentration of the analyte solution. The distance between the metal CGs should be sufficiently small to decrease parasitic capacitances associated with the surface areas between the sensors uncovered with the metal CG layer. Conversely, this distance should be sufficiently large to prevent overlapping of the depletion regions in the semiconductor due to the fringing effect and, thereby, practically eliminate possible cross-talk effects between the on-chip integrated CG-EISCAPs.

The CG-EISCAP represents a dual-gate device combining SG and CG, which are coupled with a common floating gate (FG). Thus, the FG potential (*V*_FG_) can be modulated by either SG or CG. CG has a multi-purpose function and is connected with the multiplexer by three positions (floating “F”, short-circuited “SC” and capacitively-coupled “CC”), which enables an activation or deactivation of the particular CG-EISCAP. In contrast to conventional EISCAPs, the proposed design allows independent biasing and tuning of the operating point of each CG-EISCAP sensor in the desired region of the capacitance-voltage (*C–V*) curve (accumulation, depletion or inversion) by means of applying an additional voltage on the respective CG. This way, possible device-to-device differences in the flat-band voltage of various CG-EISCAPs caused from technological factors (e.g., inhomogeneously distributed trapped charges on the floating gate or non-uniform thickness of the gate insulator) can be compensated, too. In addition, both typical characterization modes of EISCAPs, namely the *C–V* curve and the ConCap (constant-capacitance) mode response can be recorded for each sensor separately in two ways: by means of applying an AC (alternating current) voltage (a) between the RE and the rear-side contact (as for conventional EISCAP sensors) or (b) between the CG and the rear-side contact. Finally, beside the field-effect measurement setup, the proposed structure can also be used as an impedimetric sensor or as capacitively-coupled contactless electrolyte-conductivity detection (so-called C^4^D [39]) sensor.

It is worth to mention, that in contrast to on-chip integrated EISCAP arrays reported in [36], our design uses one common RE for all sensors in the array. On the other hand, a conventional Si wafer is utilized instead of a costly SOI [23,35] or an anodically bonded Si wafer [34]. In addition, the fabrication of CG-EISCAPs is easy; it requires only one photolithographic step when depositing the CG layer via a shadow mask or two photolithographic steps in case of structuring of the CG layer by lift-off process or etching. In some embodiments, the technological process steps could also include front-side contacting to the Si instead of rear-side contacting or the preparation of an on-chip integrated common pseudo-RE.

## 3. Modelling of On-Chip Integrated, Individually Addressable CG-EISCAPs

For the development of the electrical equivalent circuit and modelling of the CG-EISCAP chip in different setups, let us assume that CG-EISCAP-1 in Figure 2 is modified with receptor R1 for the detection of target analyte T1, while CG-EISCAP-2 is modified with receptor R2 for the detection of target analyte T2. CG-EISCAP-3 is unmodified and serves for pH control of the analyte solution or as reference sensor. Since generally EG-FEDs (particularly EISCAPs) are charge-sensitive devices, any specific electrochemical interaction between the immobilized receptor and target analyte (e.g., affinity reaction, DNA hybridization, local pH changes due to enzymatic reactions, etc.) that occurs at or immediately near the gate surface (within the so-called Debye length from the surface) will induce changes in the surface charge/potential of gate insulator-2 that will consequently modulate the overall capacitance of the EISCAP sensor (see e.g., recent review [15]).

The complete electrical equivalent circuit of the CG-EISCAP chip is complex and involves components associated with the resistance of the RE (*R_RE_*), resistance of the bulk solution, double-layer capacitance at the electrolyte/insulator-2 interface, capacitances of insulator-1 and insulator-2, the space-charge capacitance in the semiconductor, and resistances of the bulk semiconductor and the metal-semiconductor rear-side contact. However, as discussed in [15,40,41], for typical gate insulator films used for EISCAPs (e.g., SiO_2_, Si_3_N_4_, Al_2_O_3_, Ta_2_O_5_) and their usual thickness range (10–100 nm) as well as appropriate experimental conditions (ionic strength of the solution >0.1 mM; measurement frequencies of <1 kHz), the interferences from several components, such as double-layer capacitance and electrolyte resistance, are negligible. In addition, the resistances of bulk Si and Al-Si rear-side contact are much smaller than *R_RE_* and therefore, can be also neglected. Hence, the equivalent circuit of the individual CG-EISCAP sensor with a floating CG can be simplified as a series connection of capacitances of insulator-2, insulator-1 and the variable space-charge capacitance of the semiconductor.

Figure 3 represents the simplified equivalent circuit of the chip composed of three CG-EISCAPs. Here, *C_i*1*_*, *C_i*2*_* and *C_nsc_* (*n* = 1, 2, 3) are the capacitances of insulator-1, insulator-2 and space-charge region in the semiconductor associated with CG-EISCAP-1, CG-EISCAP-2 and CG-EISCAP-3, respectively. In this work, the surface-sensing areas (spots) of all three CG-EISCAPs are assumed to be equal and all capacitances are defined per unit surface area: *C_i_*_1_ = *ε*_1_/*d*_1_, *C_i*2*_* = *ε*_2_/*d*_2_, where *ε*_1_, *ε*_2_ and *d*_1_, *d*_2_ are permittivities and thicknesses of insulator-1 and insulator-2, respectively. The space-charge capacitances of CG-EISCAP-1, CG-EISCAP-2 and CG-EISCAP-3 are given as: *C*_1*sc*_ = *ε_s_*/*w*_1_, *C_2sc_* = *ε_s_*/*w*_2_, *C*_3*sc*_ = *ε_s_*/*w*_3_, respectively, where *ε_s_* is the permittivity of the semiconductor, and *w_1_*, *w_2_* and *w_3_* are the widths of the corresponding depletion regions. The width of the depletion region and consequently, the space-charge capacitance of each sensor will be determined—among others—by the applied voltage on the gate (in this case, SG and/or CG) and by the respective electrolyte/insulator-2 interfacial potentials. We assume that the surface areas of insulator-1 covered with metal CGs are much larger than that of metal-free areas. Therefore, the parasitic capacitance associated with the surface areas between the individual CG-EISCAPs (uncovered with metal CG layer) has not been included in the equivalent circuit.

The equivalent capacitance of the chip (*C_eq_*) is determined as:(1)$$C_{eq}=C_{1}+C_{2}+C_{3}\;$$
where *C*_1_, *C*_2_, and *C*_3_ are the overall capacitances of CG-EISCAP-1, CG-EISCAP-2 and CG-EISCAP-3, respectively, which are determined by the combination of the capacitances *C_i_*_1_, *C_i_*_2_ and *C_nsc_* in series:(2)$$\frac{1}{C_{1}}=\frac{1}{C_{i1}}+\frac{1}{C_{i2}}+\frac{1}{C_{1sc}}\;$$
(3)$$\frac{1}{C_{2}}=\frac{1}{C_{i1}}+\frac{1}{C_{i2}}+\frac{1}{C_{2sc}}$$
(4)$$\frac{1}{C_{3}}=\frac{1}{C_{i1}}+\frac{1}{C_{i2}}+\frac{1}{C_{3sc}}\;$$

As can be seen in Figure 3, the individual CG-EISCAPs in the array are still interconnected via the common Si substrate, which may result in an unwanted cross-talk between the on-chip integrated sensors (see Introduction). For example, if the chip is exposed to the solution containing both T1 and T2 target analytes, the interaction of target T1 with the immobilized receptor R1 will modulate the interfacial potential (*φ*_1_) as well as the space-charge (*C*_1*sc*_) and overall (*C*_1_) capacitance of CG-EISCAP-1. Analogously, the interaction of target T2 with the immobilized receptor R2 and/or possible pH changes will modulate the overall capacitances of CG-EISCAP-2 (*C*_2_) and CG-EISCAP-3 (*C*_3_), respectively, all resulting in a change of the equivalent capacitance, *C_eq_*, of the chip. As a consequence, such a chip is unable to selectively distinguish between particular target analytes in a multicomponent solution. However, the existence of CGs enables addressable activation/deactivation of individual CG-EISCAPs by switching the CG of each sensor in various setups, such as floating/short-circuited CG or floating/capacitively-coupled CG, which are discussed below. This feature of an addressable activation or deactivation of on-chip integrated individual CG-EISCAPs by simple electrical switching the respective CG (instead of fabricating an array of separate EISCAPs) makes the new design capable for multiplexed operation. Detection of multiple analytes is possible, eliminating cross-talk effects between the sensors in the array.

### 3.1. Setup with Floating/Short-Circuited CG

Figure 4 shows the electrical equivalent circuit and measurement setup of the chip with floating/short-circuited CG. The chip is exposed to the solution containing multiple target analytes (exemplarily, T1 and T2) and the gate voltage *V_G_* is applied to the structure via the RE to set the working points of all three sensors in the depletion region.

To detect target analyte T1 with the CG-EISCAP-1, CG1 is kept floating (CG1 is switched to position “F” of the multiplexer), while CG2 and CG3 should be short-circuited (switched to position “SC”, Figure 2) to exclude an impact of possible gate-surface potential changes of CG-EISCAP-2 and CG-EISCAP-3 on the total capacitance of the chip and, therefore, on the output signal. FG1 transfers the signal from the electrolyte/insulator-2 interface to the semiconductor in an electrostatic way: the floating gate potential of CG-EISCAP-1, *V_FG_*_1_, will follow the changes in both the gate voltage (*V_G_*) and the interfacial potential (*φ*_1_). The term *φ*_1_ can be represented as *φ*_1_
*=*
*φ*_01_
*±* Δ*φ*_1_, where *φ*_01_ is the potential at the insulator-2/electrolyte interface before the biochemical interaction of target T1 with the immobilized receptor R1 and Δ*φ*_1_ is the potential change induced via the biochemical interaction. The expression for *V_FG_*_1_ of CG-EISCAP-1 can be obtained using the capacitive voltage divider model:(5)$$V_{FG1}=\frac{C_{i2}}{C_{t1}}V_{G1-eff}\;$$
where *V_G_*_1*-eff*_ is the effective gate voltage and is given by [42,43]:(6)$$V_{G1-eff}=V_{G}-V_{op}=V_{G}-E_{ref}+\varphi_{1}-\chi_{sol}+W_{m}/q$$

Here, *V_op_* is the overall potential drop through the RE/electrolyte/insulator system, *E_ref_* is the potential of the RE relative to vacuum, *χ_sol_* is the surface-dipole potential of the solvent, *W_m_* is the metal electron work function, *q* is the elementary charge (1.6 × 10*^−^*^19^ C) and *C_t_*_1_ is the sum of all capacitances coupled to the floating node with:(7)$$C_{t1}=C_{i2}+\frac{C_{i1}C_{1sc}}{C_{i1}+C_{1sc}}\;$$

The simplified equivalent circuit corresponding to the floating/short-circuited CG setup is shown in Figure 4 (right), where the equivalent capacitance (*C_eq_*) of the chip is determined as:(8)$$C_{eq}=C_{1}+C_{2}+C_{3}=C_{i1}C_{i2}C_{1sc}/\left[C_{1sc}\left(C_{i1}+C_{i2}\right)+C_{i1}C_{i2}\right]+2C_{i2}\;$$

In general, in the presence of a series resistance (e.g., resistance of the *RE*, *R_RE_*), the measured capacitance (*C_m_*) will be given by [35,44,45]:(9)$$C_{m}=C_{eq}/\left[1+{\left(2\pi fR_{RE}C_{eq}\right)}^{2}\right]\;$$
where *f* is the measurement frequency. *C_m_* will be equal to *C_eq_*, if (2π*fR_RE_C_eq_*)^2^ << 1. Otherwise, *C_m_* will be affected by the series resistance, resulting in frequency-dependent *C–V* curves and a much smaller *C_m_* than the real capacitance of the system.

In Equation (8), all terms are constant except *C*_1*sc*_, which at a constant *V_G_* presumably will depend on the T1 concentration in solution. Thus, the chip will detect explicitly potential changes on the gate surface of CG-EISCAP-1 resulting from the interaction of T1 with R1. Although, the gate surface potential of CG-EISCAP-2 will be also altered due to the interaction of T2 with R2, this has no impact on the results of the detection of T1 with CG-EISCAP-1 (because it is short-circuited). Consequently, there are no cross-talk effects between the individual CG-EISCAP sensors in the array. Similarly, for the detection of the target analyte T2 with CG-EISCAP-2, CG2 should be held as floating, while CG1 and CG3 should be short-circuited. Finally, for the pH control with the CG-EISCAP-3, CG1 and CG2 should be short-circuited, while CG3 should be switched to position “F”.

To compare the shape of the expected *C–V* curve and the potential sensitivity of the CG-EISCAP chip and the single EISCAP (without control gate), let us determine *C_eq_* in the accumulation, depletion and inversion region, respectively. In the accumulation region (*V_G_* < 0), *C*_1*sc*_ >> *C_i_*_1_ and *C*_1*sc*_ >> *C_i_*_2_. Then, the equivalent capacitance of the chip in the accumulation region (*C_eq-acc_*) can be derived from Equation (8) as:(10)$$C_{eq-acc}=C_{i1}C_{i2}/\left(C_{i1}+C_{i2}\right)+2C_{i2}\;$$

By strong inversion, the depletion-layer width reaches a maximum, *w_m_* [46]:(11)$$w_{m}=\sqrt{\frac{4\varepsilon_{s}kTln\left(N_{A}/n_{i}\right)}{q^{2}N_{A}}}\;$$
where *k* is the Boltzmann’s constant, *T* is the temperature, *N_A_* is the density of ionized acceptors (p-Si) and *n_i_* is the electron density in the intrinsic semiconductor. The corresponding high-frequency capacitance of EISCAP-1 in the inversion range (*C*_1*inv*_) reaches its minimum. The equivalent capacitance of the chip in the inversion range (*C_eq-inv_*) can be obtained from Equation (8) by replacing *C*_1*sc*_ with *C*_1*inv*_ = *ε_s_*/*w_m_*:(12)$$C_{eq-inv}=C_{i1}C_{i2}C_{1inv}/\left[C_{1inv}\left(C_{i1}+C_{i2}\right)+C_{i1}C_{i2}\right]+2C_{i2}$$

Typically, *C*_1*inv*_ << *C_i_*_1_ and *C*_1*inv*_ << *C_i_*_2_, hence, Equation (12) can be simplified as
(13)$$C_{eq-inv}=C_{1inv}+2C_{i2}$$

For biochemical sensor applications, more interestingly in the depletion region, the space-charge capacitance in the semiconductor and, therefore, the overall capacitance of the chip depends on both the gate voltage and the interfacial potential. In Equation (8) for *C_eq_*, the only variable term is the space-charge capacitance (*C*_1*sc*_), which can be deduced from the expression for the depletion capacitance of a MOS (metal-oxide-semiconductor) capacitor (*C_scMOS_*) [46]:(14)$$C_{scMOS}=\frac{1}{\sqrt{\frac{1}{C_{i}^{2}}+\frac{2\left(V_{G}-V_{FB}\right)}{qN_{A}\varepsilon_{s}}}-\frac{1}{C_{i}}}$$

For this, the flat-band voltage *V_FB_* (the externally applied voltage needed to make energy bands in the semiconductor flat from bulk to the surface and the net charge density in the semiconductor to zero) and the gate-insulator capacitance (*C_i_*) of the MOS structure are replaced by the flat-band voltage (*V_fb_*_1_) and series capacitances *C_i_*_1_ and *C_i_*_2_ (*C_i_* = *C_i_*_1_*C_i_*_2_/(*C_i_*_1_+*C_i_*_2_)) of the CG-EISCAP-1, respectively:(15)$$C_{1sc}=\frac{1}{\sqrt{{\left(\frac{C_{i1}+C_{i2}}{C_{i1}C_{i2}}\right)}^{2}+\frac{2\left(V_{G}-V_{fb1}\right)}{qN_{A}\varepsilon_{s}}}-\frac{C_{i1}+C_{i2}}{C_{i1}C_{i2}}}$$

Generally, the flat-band voltage of the EISCAP is given by [47]:(16)$$V_{fb1}=E_{ref}-\varphi_{1}+\chi_{sol}-\frac{W_{s}}{q}-\frac{Q_{i}+Q_{ss}}{C_{i}}$$
where *W_s_* is the silicon electron work function, and *Q_i_* and *Q_ss_* are the charges located in the oxide and the surface and interface states, respectively. By assuming that *Q_i_* and *Q_ss_*, and the charge at the floating gate are zero and grouping analyte-concentration independent potentials in *V_ip_* = *E_ref_* + *χ_sol_* − *W_s_*/*q*, the expression (16) for the flat-band voltage can be simplified as:(17)$$V_{fb1}=V_{ip}-\varphi_{1}$$

By substituting expressions (15) and (17) into Equation (8), we obtain the following equation for the equivalent capacitance of the chip in the depletion region, *C_eq-dep_*:(18)$$C_{eq-dep}=\frac{1}{\sqrt{{\left(\frac{C_{i1}\;+\;C_{i2}}{C_{i1}C_{i2}}\right)}^{2}+\frac{2\left(V_{G}\;-\;V_{ip}\;+\;\varphi_{1}\right)}{qN_{A}\varepsilon_{s}}}}+2C_{i2}$$

At a constant *V_G_*, all terms in Equation (18) can be considered as constant except for *φ*_1_, which is analyte-concentration dependent. The combination of Equations (10), (12) and (18) gives the complete description of the *C–V* curve. The sensitivity (*S**_φ_*) of the chip to surface potential changes induced by the receptor-target analyte interaction onto the CG-EISCAP-1 surface can be obtained by differentiation of *C_eq-dep_* with respect to *φ*_1_:(19)Sφ=|dCeq-depdφ1|=1qεsNa[(Ci1 + Ci2Ci1Ci2)2+2(VG − Vip + φ1)qεsNA]32

The analysis of Equations (10), (13), (18) and (19) reveals that the *C–V* curve of the chip will have the same shape as for a single conventional EISCAP sensor (without control gate) with the same stacked double-gate insulators and gate surface area as the CG-EISCAP-1, but will be shifted parallel along the capacitance axis with the amount of 2*C_i_*_2_, as shown in Figure 5. The overall capacitance and output signals of other CG-EISCAPs will remain unchanged. The chip combining an array of CG-EISCAPs will respond in exactly the same manner as the single conventional EISCAP sensor. Therefore, no loss in sensitivity of the CG-EISCAP chip to surface potential changes in comparison with a single EISCAP sensor will be observed. Similar expressions can be obtained in the case of measurements with the CG-EISCAP-2 and the CG-EISCAP-3 sensors. Equations (18) and (19) describe the equivalent capacitance in the depletion region and potential sensitivity of the CG-EISCAP chip without defining the origin of the potential generation at the analyte/insulator-2 interface. If the applied gate voltage, *V_G_*, is fixed, the only variable component is the interfacial potential *φ*, which is analogous to the effect of applying an additional voltage to the gate. The sensitivity of the chip to analyte concentration variations will be determined by the particular mechanism of the interfacial potential generation (e.g., pH or ion-concentration change, antibody-antigen affinity reaction, DNA hybridization, enzymatic reactions, etc.) and many other experimental factors (e.g., effective charge of the target analyte, distance of bound analyte charge from the gate surface, density of receptors, buffer capacity and ionic strength of the sample, and so on). Therefore, corresponding expressions for the analyte sensitivity, derived from other kinds of EG-FEDs, are fully transferable to CG-EISCAPs. For example, the pH sensitivity of such CG-EISCAPs can be determined as changes of the interfacial potential (*φ*) in response to a change in the bulk pH [48]:(20)$$\frac{\delta \varphi }{\delta pH}=-2.3\frac{kT}{q}\alpha \;$$
(21)$$\mathrm{with}\;\alpha =\frac{1}{\left(2.3\;kTC_{DL}/q^{2}\beta_{int}\right)+1}\;$$

Here, *α* is a dimensionless sensitivity parameter, varying between 0 and 1, *β_int_* is the surface intrinsic buffer capacity that characterizes the ability of the oxide surface to release or bind protons, and *C_DL_* is the double-layer capacitance.

### 3.2. Setup with Floating/Capacitively-Coupled CGs

Figure 6 shows the equivalent circuit and measurement setup of the chip with floating/capacitively-coupled CGs. First, the CGs of all three sensors are floating (switched to position “F”, Figure 2) and the working points of all three sensors are fixed in the depletion region by applying the gate voltage, *V_G_*, via the RE. To detect target analyte T1 with the CG-EISCAP-1, CG1 is kept floating, while CG2 and CG3 are capacitively coupled (switched to position “CC”, Figure 2) to the floating gates of FG2 and FG3 via external (or technologically on-chip integrated) capacitances *C_CG_*_2_ and *C_CG_*_3_, respectively. In this setup, in addition to SG, the capacitively coupled CG2 and CG3 can also be used to modulate the space-charge capacitances of the CG-EISCAP-2 and the CG-EISCAP-3. The floating gate voltage of the CG-EISCAP-2 (*V_FG_*_2_) or the CG-EISCAP-3 (*V_FG_*_3_) is established by a weighted sum of the two input voltages, namely, the effective gate voltage (*V_G_*_2*-eff*_ or *V_G_*_3*-eff*_) and the control gate voltage (*V_CG_*_2_ or *V_CG_*_3_). The expressions for *V_FG_*_2_ and *V_FG_*_3_ can be obtained by taking into account that each weight is determined by the capacitance of its input normalized by the total capacitance (*C_t_*_2_ or *C_t_*_3_) coupled to the floating node:(22)$$V_{FG2}=\frac{C_{i2}V_{G2-eff}+C_{CG2}V_{CG2}}{C_{2t}}\;$$
(23)$$V_{FG3}=\frac{C_{i2}V_{G3-eff}+C_{CG3}V_{CG3}}{C_{3t}}\;$$
(24)$$C_{t2}=C_{i2}+\frac{C_{i1}C_{2sc}}{C_{i1}+C_{2sc}}+C_{CG2}$$
(25)$$C_{t3}=C_{i2}+\frac{C_{i1}C_{3sc}}{C_{i1}+C_{3sc}}+C_{CG3}$$
where *V_G_*_2*-eff*_ and *V_G_*_3*-eff*_ are determined by Equation (6) by replacing *φ*_1_ with *φ*_2_ or *φ*_3_, respectively.

To exclude an impact of possible gate-surface potential changes of the CG-EISCAP-2 and the CG-EISCAP-3 on the total capacitance of the chip, the CG-EISCAP-2 and the CG-EISCAP-3 have to be deactivated by switching to position “CC” (Figure 6) and applying a voltage on CG2 and CG3. The CG-EISCAP-2 and the CG-EISCAP-3 can be deactivated by shifting the operation point either to the accumulation or strong inversion state, where the overall capacitance of these sensors is independent of the gate voltage (*V_G_*) or respective interfacial potentials (*φ*_2_, *φ*_3_). In the following, exemplarily, the expressions for the equivalent capacitance of the CG-EISCAP chip are obtained by assuming that the CG-EISCAP-2 and the CG-EISCAP-3 are deactivated by shifting their operation point in the strong inversion region.

By assuming that *C_CG_*_2_ = *C_CG_*_3_ = *C_CG_* and *C*_2*sc*_ = *C*_3*sc*_ = *C_inv_*, the overall capacitance of the CG-EISCAP-2 (*C*_2_) or the CG-EISCAP-3 (*C*_3_) in the inversion region is given by:(26)$$C_{2}=C_{3}=\frac{C_{i2}\left[C_{CG}\left(C_{i1}+C_{inv}\right)+C_{i1}C_{inv}\right]}{\left(C_{i2}+C_{CG}\right)\left(C_{i1}+C_{inv}\right)+C_{i1}C_{inv}}$$

Typically, *C_inv_* << *C_i_*_1_, hence, Equation (26) can be simplified as:(27)$$C_{2}=C_{3}=\frac{C_{i2}\left(C_{CG}+C_{inv}\right)}{C_{i2}+C_{CG}+C_{inv}}$$

The equivalent capacitance of the chip in the accumulation, inversion and depletion regions can be derived from Equations (10), (13) and (18) by replacing *C_i_*_2_ by *C*_2_, Equation (27):(28)$$C_{eq-acc}=\frac{C_{i1}C_{i2}}{C_{i1}+C_{i2}}+2\frac{C_{i2}\left(C_{CG}+C_{inv}\right)}{C_{i2}+C_{CG}+C_{inv}}$$
(29)$$C_{eq-inv}=C_{1inv}+2\frac{C_{i2}\left(C_{CG}+C_{inv}\right)}{C_{i2}+C_{CG}+C_{inv}}$$
(30)$$C_{eq-dep}=\frac{1}{\sqrt{{\left(\frac{C_{i1}\;+\;C_{i2}}{C_{i1}C_{i2}}\right)}^{2}+\frac{2(V_{G}\;-\;V_{ip}\;+\;\varphi_{1})}{qN_{A}\varepsilon_{s}}}}+2\frac{C_{i2}\left(C_{CG}+C_{inv}\right)}{C_{i2}+C_{CG}+C_{inv}}$$

Note, since in Equation (30) the inversion capacitance, *C_inv_*, of the high-frequency *C–V* curve is independent of the gate voltage or interfacial potentials, the sensitivity (*S**_φ_*) of the chip using the setup with floating/capacitively-coupled CG will be defined by the same Equation (19) as for the setup with floating/short-circuited CG. Thus, the chip combining an array of CG-EISCAPs will respond only to surface potential changes induced by the receptor-target analyte interactions occurring onto the surface of the CG-EISCAP-1 and in exactly the same manner as the single conventional EISCAP sensor. However, the *C–V* curve will be shifted parallel along the capacitance axis with the amount of 2*C*_2_ (Equation (27)), as shown in Figure 5 (black curve). Similar expressions can be obtained in the case of detection with the CG-EISCAP-2 (CG2 is floating, while CG1 and CG3 are capacitively coupled) or the CG-EISCAP-3 (CG3 is floating and CG1 and CG2 are capacitively-coupled).

## 4. Simulation Results

We simulated *C–V* curves of the CG-EISCAP chip in floating/short-circuited and floating/capacitively-coupled arrangements and compared them with *C–V* curves of a single EISCAP using Python 3.8 simulation software (Python Software Foundation). In addition, the potential-sensitivity of the CG-EISCAP chip as a function of *V*_G_, *d*_1_ and *N*_A_ has been calculated. The simulation parameters are: *ε*_1_ = 3.9 (SiO_2_); *d*_1_ = 10–60 nm, *ε*_2_ = 25 (Ta_2_O_5_), *d*_2_ = 30 and 60 nm, *ε*_s_ = 11.7 (Si), *N_A_* = 10^14^–10^16^ cm^−^^3^, *n_i_* = 1.5 × 10^10^ cm^−^^3^, CG-EISCAP sensor area of *A* = 3 mm × 3 mm = 9 mm^2^, *q* = 1.6 × 10^−^^19^ C, *T* = 300 K, *k* = 1.38 × 10^−^^23^ J/K, *V_ip_* = 0 V, *V_G_* = −0.45–1 V, *C_CG_* = 50 nF, *φ*_01_ = 0 V, *φ*_1_ = −0.05, 0 and 0.05 V.

Figure 7 illustrates *C–V* curves of the single EISCAP and the CG-EISCAP chip in floating/short-circuited and floating/capacitively-coupled setups simulated at different values of *φ*_1_.

The *C–V* curves of the CG-EISCAP chip in floating/short-circuited and floating/capacitively-coupled setups were calculated for the accumulation, inversion and depletion regions using Equations (10), (12), (18), (28), (29) and (30), respectively. The *C–V* curves of the single EISCAP were simulated using Equations (10), (12) and (18) without the second term (2*C_i_*_2_). To depict the course of the equivalent capacitance of the CG-EISCAP chip and the single EISCAP, also in the transition range from depletion to accumulation region, all *C–V* curves in Figure 7 were extrapolated (dotted curves).

As predicted in Section 3.1, these *C–V* curves have the same shape, independent of the CG-EISCAP chip or the single EISCAP setup. However, in comparison with the *C–V* curves of the single EISCAP, the *C–V* curves of the CG-EISCAP chip are shifted along the capacitance axis in the direction of larger capacitance values due to the additional parallel constant capacitances of the other sensors in the array. The amount of these shifts is defined by the second term in Equations (10) and (28) (see also Figure 5). As expected, at a constant *C_eq_*, the *C*–*V* curves are also shifted along the voltage axis. The direction and amount of these shifts (Δ*V_G_*, see top *C*–*V* curves in Figure 7) depend on the sign and amplitude of additional potential changes induced by any biochemical interaction on the sensor surface: Δ*V_G_* = ±Δ*φ*_1_. In case of a p-type EISCAP, an additional positive potential generated by the biochemical interactions on the EISCAP surface will lead to an increase in the width of the depletion layer (correspondingly, the depletion capacitance decreases). As a consequence, the overall capacitance of the CG-EISCAP chip or the single EISCAP will also decrease, resulting in a shift of the *C–V* curve towards more negative (less positive) gate voltages (Figure 7, blue curve).

Conversely, an additional negative potential generated by the biochemical interaction on the EISCAP surface will decrease the width of the depletion layer in the Si and consequently, increase the depletion capacitance. The overall capacitance of the CG-EISCAP chip or the single EISCAP will also increase, resulting in a shift of the *C*–*V* curve in the direction of more positive (less negative) gate voltages (Figure 7, red curve). Such shifts of the *C*–*V* curve along the voltage axis upon biochemical interaction was observed in many experiments on conventional single EISCAP-based pH sensors or biosensors (e.g., [17,19,20,24,26]). Often, these sensors work in the ConCap mode, by which gate-surface potential shifts induced upon biochemical interactions can directly be determined from the dynamic sensor response (see [15] and references therein).

Figure 8 shows the calculated curves of the sensitivity of the CG-EISCAP chip on the gate voltage (a), the thickness of the insulator-1 and insulator-2 (b) and the doping concentration (c). With increasing gate voltage, the sensitivity of the CG-EISCAP chip to surface potential changes induced by the receptor-target analyte interaction is decreased. Maximum sensitivity will be achieved at the inflection point of the *C–V* curve, which corresponds to the flat-band condition as it has been discussed in [49]. In the transition range from depletion to accumulation region (i.e., at gate voltages of *V_G_* < *V_fb_*_1_), the sensitivity of the CG-EISCAP chip will again decrease (not shown), similar to conventional EISCAPs. As expected, the calculations, depicted in Figure 8b, show that the sensitivity is increased with decreasing the layer thicknesses of both insulator-1 and insulator-2. This is due to the increase in the *C_eq-acc_/C_eq-inv_* ratio and the steepness of the *C–V* curve in the depletion region.

Finally, Figure 8c illustrates the dependence of the sensitivity on the doping concentration (*N_A_*) at different thicknesses of insulator-1 (*d*_1_). The sensitivity is increased with increasing *N_A_*, reaching its maximum value, and is decreased by further increase in *N_A_*.

Such a course of *S**_φ_* curves may be explained by taking into consideration the ratio:(31)$$R={\left(\frac{C_{i1}+C_{i2}}{C_{i1}C_{i2}}\right)}^{2}/[\frac{2(V_{G}-V_{ip}+\varphi_{1})}{q\varepsilon_{s}V_{A}}]$$
in Equation (19). If *R* << 1, *S**_φ_*~*N_A_*^1/2^ and the sensitivity is increased with increasing the doping concentration. Conversely, if *R* >> 1, the *S**_φ_*~*N_A_***^−^**^1^ and the sensitivity is decreased with increasing the doping concentration. The maximum sensitivity value and its position along the *N_A_*-axis depends on *d*_1_. At a constant *V_G_* and *φ*_1_, with decreasing *d*_1_, the maximum sensitivity is increased and its position is shifted towards higher *N_A_* values.

## 5. Conclusions

Multiplexed biochips for multianalyte detection have been increasingly recognized as powerful tools in many fields of application, including point-of-care diagnostics and personalized medicine. In this work, a new design for an array of on-chip integrated, individually electrically addressable CG-EISCAPs for a multiplexed (quasi)simultaneous detection of multiple analytes is presented. In comparison with conventional EISCAPs, CG-EISCAPs have a supplemental control gate in addition to their sensing gate, which enables the activation or deactivation of individual CG-EISCAPs inside the array, thus (practically) eliminating possible cross-talk effects between the sensors.

The new designed CG-EISCAP chip was modelled for two setups (floating/short-circuited CG and floating/capacitively-coupled CG). To validate the equivalent-circuit model of the CG-EISCAP chip, the capacitance-voltage curves were simulated for different setups and compared with that of a single EISCAP sensor (without CG). The simulation results reveal that the chip combining an array of CG-EISCAPs will respond in exactly the same manner as the single EISCAP sensor, without loss in sensitivity. Additional to the *C–V* curves, the sensitivity of the CG-EISCAP chip to surface potential changes induced by biochemical reactions was simulated and the impact of different parameters such as the gate voltage, the insulator thickness and the doping concentration on the sensitivity has been discussed.

In conclusion, the results achieved in this work underline a great potential of CG-EISCAPs as an alternative transducer structure for the realization of multiplexed biochips for (quasi)simultaneous detection of multiple analytes without additional process complexity and with numerous possible applications. Although in this work, an array combining three CG-EISCAPs was modelled, the proposed approach may be extended to CG-EISCAP chips consisting of *N* sensors, as well as to other kinds of EIS-based biochemical sensors (e.g., light-addressable potentiometric sensors [50]).

## Figures and Tables

**Figure 1 sensors-21-06161-f001:**
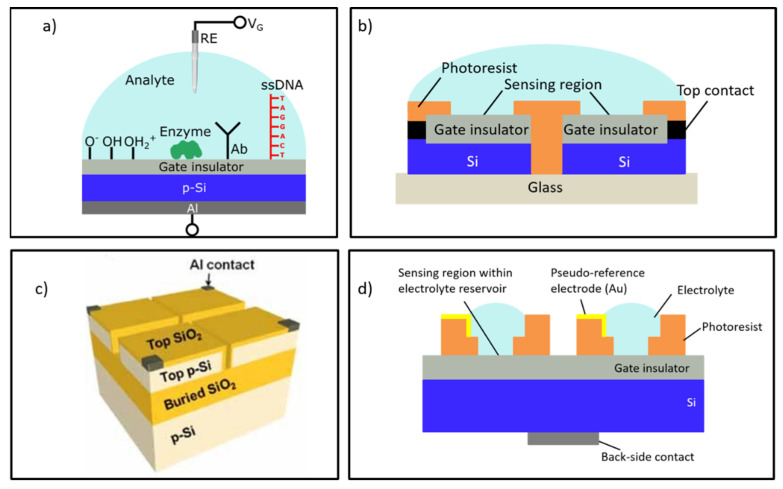
(**a**) Schematic structure of a conventional single EISCAP biochemical sensor with different receptor functionalities (reproduced from [15], open access publication under CC BY license); (**b**) layout of an EISCAP sensor array fabricated on a Si wafer anodically bonded to the glass substrate; (**c**) schematic of a chip combining a 2 × 2 array of nanoplate EISCAPs prepared on a SOI substrate (reproduced from [23] with permission from John Wiley and Sons); (**d**) design of an array of EISCAPs separated via the electrolyte reservoirs fabricated on the gate surface (schematically). RE: reference electrode, *V_G_*: gate voltage, Ab: antibody, ssDNA: single-strand deoxyribonucleic acid.

**Figure 2 sensors-21-06161-f002:**
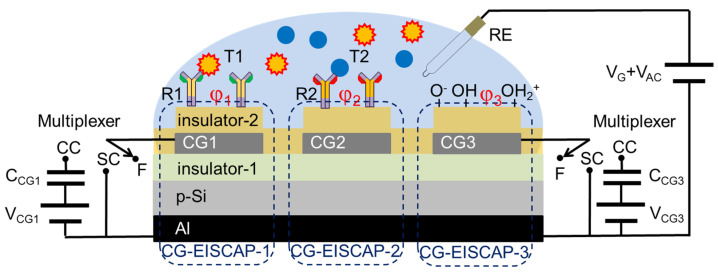
Schematic structure of the designed sensor chip for detecting multiple analytes, combining an array of three individually addressable CG-EISCAPs and measurement setup. RE: reference electrode (sensing gate, SG); *V_G_*: gate voltage; *V_AC_*: alternating current voltage; Al: rear-side contact; CG1, CG2 and CG3: control gates; position “F”: floating; position “SC”: short-circuited; position “CC”: capacitively-coupled; *C*_CG1_ and *C*_CG3_: control gate capacitances; *V*_CG1_ and *V*_CG3_: voltage applied to the control gate; R1 and R2: receptors; T1 and T2: target species to be detected; *φ*_1_, *φ*_2_ and φ_3_: potential at the insulator-2/electrolyte interface related to CG-EISCAP-1, CG-EISCAP-2 and CG-EISCAP-3, respectively. For better visibility, the multiplexer for CG-EISCAP-2 is not shown.

**Figure 3 sensors-21-06161-f003:**
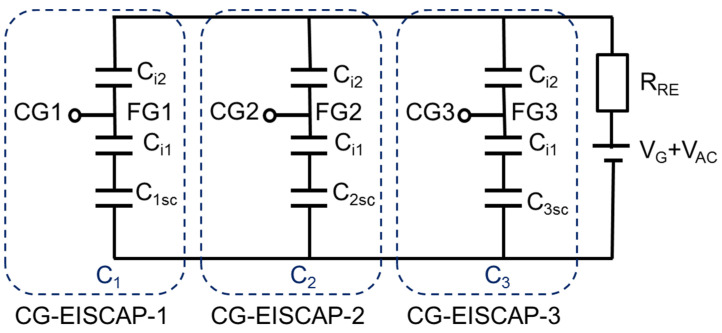
Electrical equivalent circuit of the chip consisting of three CG-EISCAPs. *C_i_*_1_ and *C_i_*_2_: capacitances of insulator-1 and insulator-2, respectively; *C*_1*sc*_, *C*_2*sc*_ and *C*_3*sc*_: capacitances of the space-charge region in the semiconductor associated with CG-EISCAP-1, CG-EISCAP-2 and CG-EISCAP-3, respectively; CG1, CG2 and CG3: control gates; *V_G_*: gate voltage; *V_AC_*: alternating current voltage; *R_RE_*: resistance of RE; FG1, FG2 and FG3: floating gates; *C*_1_, *C*_2_ and *C*_3_: overall capacitances of CG-EISCAP-1, CG-EISCAP-2 and CG-EISCAP-3, respectively.

**Figure 4 sensors-21-06161-f004:**
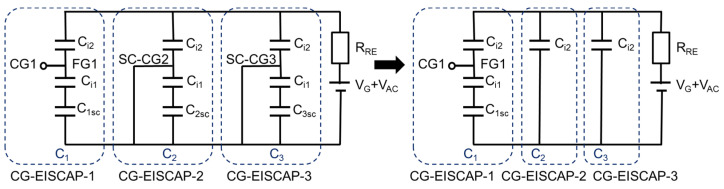
Equivalent circuit of the chip in floating/short-circuited CG setup. SC-CG2 and SC-CG3: short-circuited CG2 and CG3, respectively. All other terms are described in Figure 3.

**Figure 5 sensors-21-06161-f005:**
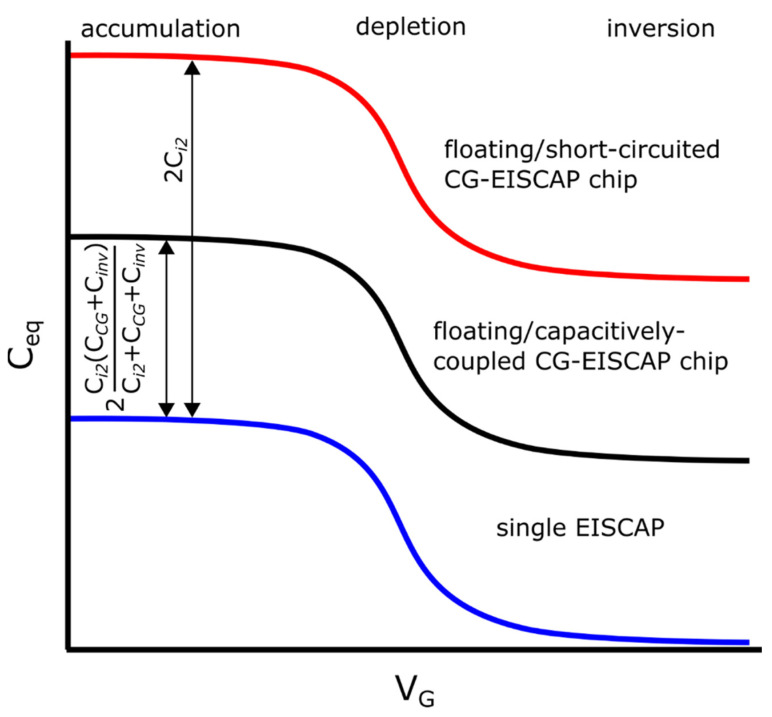
*C–V* curves of a single EISCAP sensor (blue), a floating/short-circuited (red) and a floating/capacitively-coupled (black) CG-EISCAP chip (schematically).

**Figure 6 sensors-21-06161-f006:**
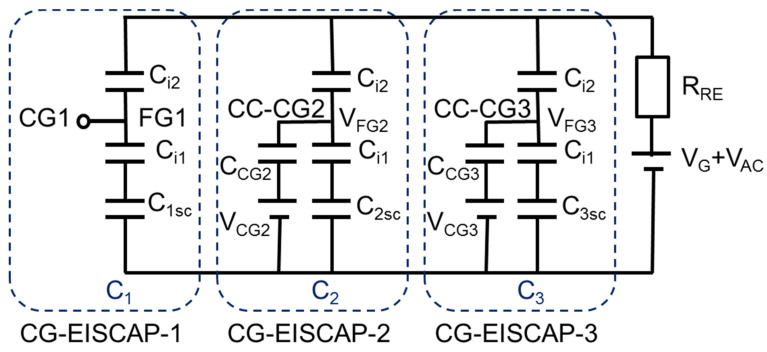
Equivalent circuit and measurement setup of the chip with a floating/capacitively-coupled CG setup. *V_CG_*_2_ and *V_CG_*_3_: voltage applied to CG2 and CG3, respectively; *V_FG_*_2_ and *V_FG_*_3_: voltage on the floating gate FG2 and FG3, respectively; *C_CG_*_2_ and *C_CG_*_3_: control-gate capacitances; CC-CG2 and CC-CG3: capacitively-coupled control gate of CG-EISCAP-2 and CG-EISCAP-3, respectively.

**Figure 7 sensors-21-06161-f007:**
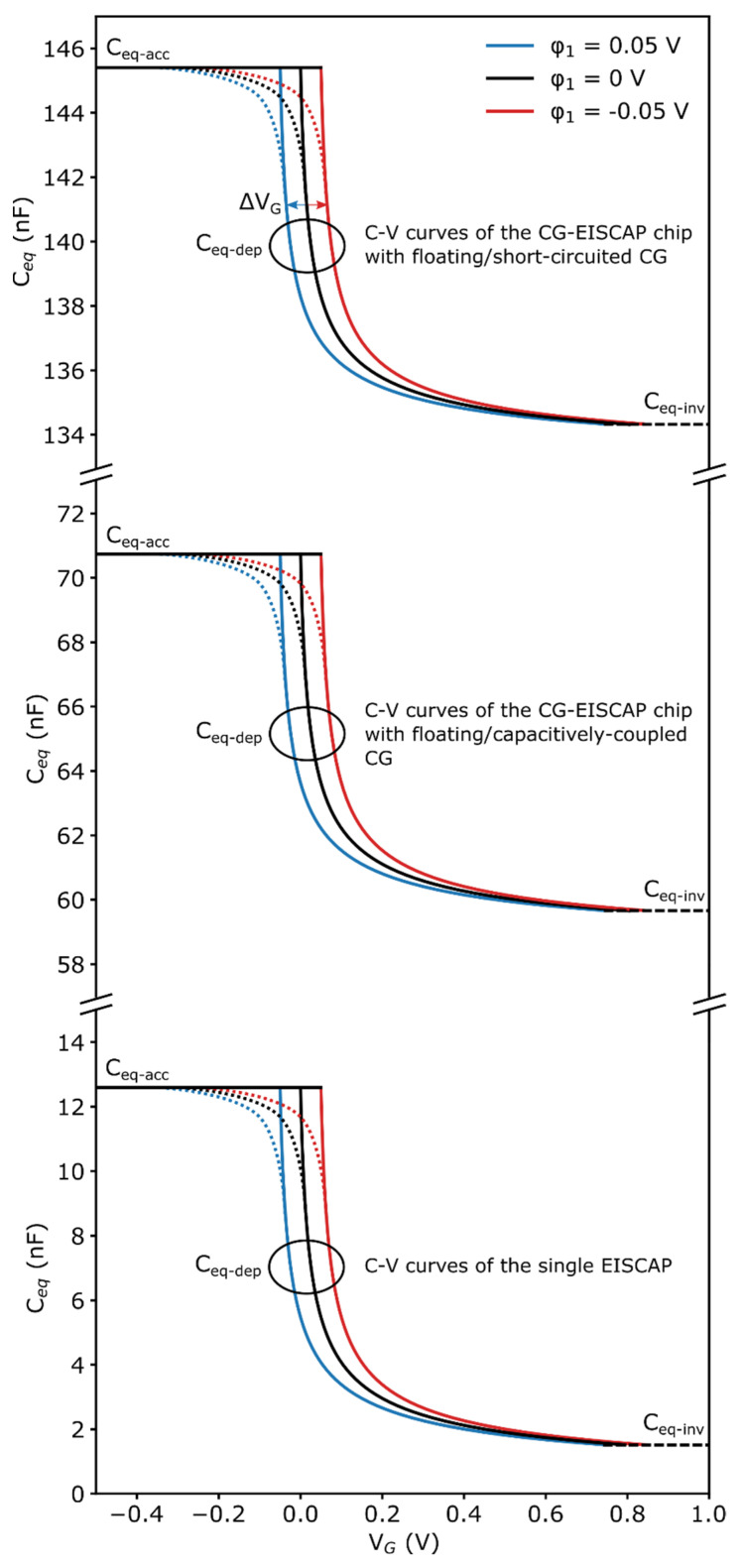
*C–V* curves of the single EISCAP and the CG-EISCAP chip in floating/short-circuited and floating/capacitively-coupled setups simulated at different values of *φ*_1_ = −0.05, 0 and 0.05 V. The dotted curves depict the extrapolated course of the overall equivalent capacitance of the CG-EISCAP chip and the single EISCAP in the transition region from depletion to accumulation. Simulation parameters: *d*_1_ = 20 nm, *d*_2_ = 30 nm, *N_A_* = 2.76 × 10^15^ cm^−3^ (that corresponds to the resistivity of Si wafer of 5 Ω∙cm). Δ*V_G_*: shift of the *C*–*V* curves induced by the biochemical interactions on the sensor surface.

**Figure 8 sensors-21-06161-f008:**
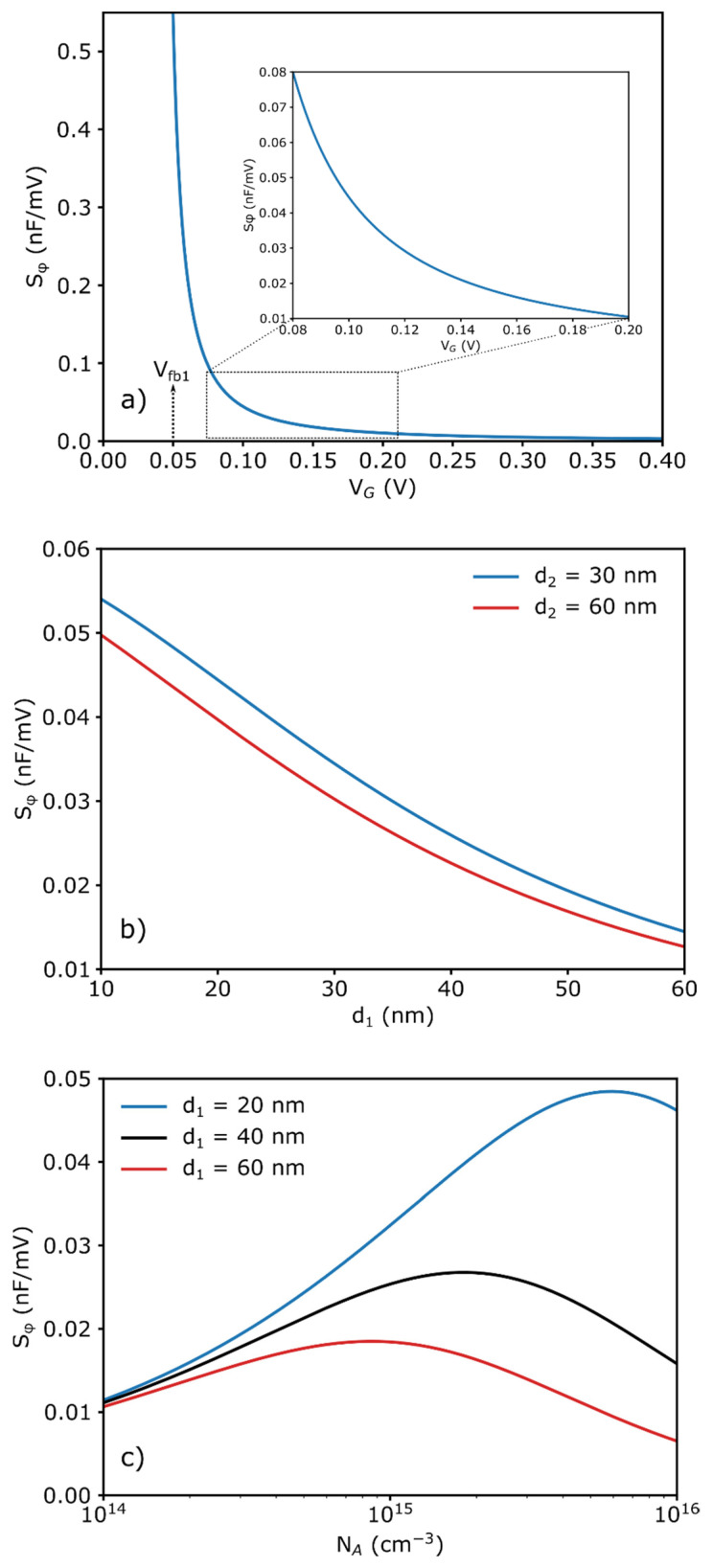
(**a**) Sensitivity of the CG-EISCAP chip as a function of gate voltage, (**b**) thickness of insulator-1 and insulator-2 and (**c**) doping concentration. Simulation parameters for (**a**): *d*_1_ = 20 nm, *d*_2_ = 30 nm, *N_A_* = 2.76 × 10^15^ cm^−^^3^, *φ*_1_ = −0.05 V; for (**b**): *N_A_* = 2.76 × 10^15^ cm^−^^3^, *V_G_* = 0.1 V, *φ*_1_ = −0.05 V; for (**c**): *d_2_* = 30 nm, *V_G_* = 0.1 V, *φ*_1_ = −0.05 V.

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
