# Peer review of "An Array of On-Chip Integrated, Individually Addressable Capacitive Field-Effect Sensors with Control Gate: Design and Modelling"

_sensors, 2021, doi:10.3390/s21186161_

Round 1

Reviewer 1 Report

This Manuscript reported a new simple process design for individually electrically addressable EISCAPs (CG-EISCAP) with additional control gates for arrays integrated on chip. The author presented a simulation of the sensitivity of the CG-EISCAP chip to changes in surface potential caused by biochemical reactions. In general, the manuscript is not systematic enough and lack of work meaning and practical application direction. Therefore, I recommend this manuscript should be rejected with the following comments.

  1. This manuscript lacks specific process schematic, please supplement corresponding process support and briefly explain.
  2. Is the process optimization universally adaptable if the model(such as Fig. 1a) is replaced?
  3. There are too many formulas and lack of in-depth parameter determination and analysis optimization. Please give the key influencing factors.
  4. The conclusion is too complicated to highlight the main point, so it is suggested to cut the length and summarize again.

Author Response

This manuscript reported a new simple process design for individually electrically addressable EISCAPs (CG-EISCAP) with additional control gates for arrays integrated on chip. The authors presented a simulation of the sensitivity of the CG-EISCAP chip to changes in surface potential caused by biochemical reactions. In general, the manuscript is not systematic enough and lack of work meaning and practical application direction. Therefore, I recommend this manuscript should be rejected with the following comments.

  1. This manuscript lacks specific process schematic, please supplement corresponding process support and briefly explain.

Answer 1: Unfortunately, the reviewer has not specified which kind of process schematic he is missing. We suspect that the comment relates to technological process steps for the preparation of CG-EISCAPs. The aim of this paper was, however, the modelling and simulation of new designed on-chip integrated individually addressable EISCAPs for multiplexed detection of multiple analytes. An additional description of technological processes of fabrication of a CG-EISCAP chip could increase the length of the manuscript significantly. Therefore, we presented the schematic layer  structure of the chip (exemplarily combining three CG-EISCAPs, Fig. 2). In comparison to conventional EISCAPs, which are based on an electrolyte-insulator-semiconductor system, the new designed CG-EISCAPs are composed of an electrolyte-insulator-metal-insulator-semiconductor structure, where the patterned metal layer between the stacked two insulators plays the role of the particular CG (see also Section 2, page 3-4). Thus, the difference between the technological process steps of fabrication of conventional EISCAPs (which are described in many previous papers, including our papers) and CG-EISCAPs is the deposition of the additionally patterned metal layer. For more details concerning the set-up and fabrication of conventional EISCAPs, see also [15] in the reference list of the paper.

  1. Is the process optimization universally adaptable if the model (such as Fig.1a) is replaced?

Answer 2: Yes, the design and technological process steps are adaptable for different modifications of EISCAPs with different kinds of receptor layers as well as for LAPS devices. Nonetheless, this paper is not dealing with process optimization rather than to theoretically model and design addressable EISCAPs.

  1. There are too many formulas and lack of in-depth parameter determination and analysis optimization. Please give the key influencing factors.

Answer 3: The submitted paper is intended to give both the expert user (academic researchers, industry professionals) and graduate students a detailed view into the new designed EISCAPs, which have been not yet discussed in literature. The formulas are necessary to understand the described setups as well as to perform the mathematical simulations. The influence of the most important, different factors (gate voltage, doping concentration, thicknesses of gate insulators) on the sensitivity of the chip have been simulated and discussed in the Section “Simulation results”.

  1. The conclusion is too complicated to highlight the main point, so it is suggested to cut the length and summarize again.

Answer 4: The conclusions highlight the design and equivalent circuit of the CG-EISCAP chip, simulation results as well as the impact of different parameters on the sensitivity of the chip. According to the reviewer’s suggestion, we shortened the section “Conclusions”.

Reviewer 2 Report

The authors report the development of a new design for an array of on-chip integrated, individually electrically addressable CG-EISCAPs. A supplemental control gate in addition to their sensing gate enables the activation or deactivation of individual CG-EISCAPs inside the array, thus practically eliminating possible cross-talk effects between the sensors. The simulated C-V curves indicate that the chip responds in the same manner as the single EISCAP sensor, without losing sensitivity. The sensitivity to surface potential changes induced by biochemical reactions was also simulated. In summary,  CG-EISCAPs are promising for simultaneous detection of multiple analytes without additional process complexity. I think the paper can be accepted in the current form.

Author Response

The authors report the development of a new design for an array of on-chip integrated, individually electrically addressable CG-EISCAPs. A supplemental control gate in addition to their sensing gate enables the activation or deactivation of individual CG-EISCAPs inside the array, thus practically eliminating possible cross-talk effects between the sensors. The simulated C-V curves indicate that the chip responds in the same manner as the single EISCAP sensor, without losing sensitivity. The sensitivity to surface potential changes induced by biochemical reactions was also simulated. In summary, CG-EISCAPs are promising for simultaneous detection of multiple analysis without additional process complexity. I think the paper can be accepted in the current form.

Answer: No comments are given by the reviewer to modify the article. Thanks a lot for the encouraging evaluation!

Reviewer 3 Report

The authors model a capacitive field-effect sensor using individually addressable EISCAPs to tackle a critical challenge in multianalyte detection: efficient multiplexing without crosstalk between distinct sensor elements functionalized with various receptors targeting a multicomponent media. The authors provide a nice background on prior work in the field to achieve on-chip integration of multiple EISCAPs, and how their alternative structure incorporating  may avoid the observed shortcomings. The manuscript is pleasant to read and convincing. Many questions that arise during the reading are answered in the next paragraphs. The authors are forthright with the limitations of their design, most notably, the inability to operate multiple sensors at the same time for simultaneous detection of multiple targets using more than one EISCAP. The manuscript is well organized and clearly written, and I recommend publication.

Author Response

The authors model a capacitive field-effect sensor using individually addressable EISCAPs to tackle a critical challenge in multianalyte detection: efficient multiplexing without crosstalk between distinct sensor elements functionalized with various receptors targeting a multicomponent media. The authors provide a nice background on prior work in the field to achieve on-chip integration of multiple EISCAPs, and how their alternative structure incorporating may avoid the observed shortcomings. The manuscript is pleasant to read and convincing. Many questions that arise during the reading are answered in the next paragraphs. The authors are forthright with the limitations of their design, most notably, the inability to operate multiple sensors at the same time for simultaneous detection of multiple targets using more than one EISCAP. The manuscript is well organized and clearly written, and I recommend publication.

Answer: No comments are given by the reviewer to modify the article. We are pleased about this assessment of our article and like to thank the reviewer for his positive evaluation.

Round 2

Reviewer 1 Report

accept